# The Effect of Mobile Lifestyle Intervention Combined with High-Protein Meal Replacement on Liver Function in Patients with Metabolic Dysfunction-Associated Steatotic Liver Disease: A Pilot Randomized Controlled Trial

**DOI:** 10.3390/nu16142254

**Published:** 2024-07-13

**Authors:** Eunbyul Cho, Sunwoo Kim, Soonkyu Kim, Ju Young Kim, Hwa Jung Kim, Yumi Go, Yu Jung Lee, Haesol Lee, Siye Gil, Sung Kwon Yoon, Keonho Chu

**Affiliations:** 1Department of Family Medicine, Seoul National University Bundang Hospital, Seongnam-si 13620, Republic of Korea; starsilver78@gmail.com; 2Health Promotion Center, Seoul Bumin Hospital, Seoul 07590, Republic of Korea; minerva87sun@naver.com; 3Bionutrion Corp., Seoul 06097, Republic of Korea; ceo@bionutrion.kr (J.Y.K.); koyumi320@bionutrion.kr (Y.G.); jamie@bionutrion.kr (Y.J.L.); hayley@bionutrion.kr (H.L.); hipisiye@bionutrion.kr (S.G.); sky7777@bionutrion.kr (S.K.Y.); chuuuu@bionutrion.kr (K.C.); 4Department of Clinical Epidemiology and Biostatistics, Asan Medical Center, University of Ulsan College of Medicine, Seoul 05505, Republic of Korea; rsvp@amc.seoul.kr

**Keywords:** metabolic dysfunction-associated steatotic liver disease, meal replacement, digital health technologies, mobile intervention, obesity

## Abstract

While many studies have explored dietary substitutes and mobile apps separately, a combined approach to metabolic dysfunction-associated steatotic liver disease (MASLD) has not been investigated. This study evaluated short-term mobile interventions coupled with partial meal replacement in patients with MASLD. Sixty adults with MASLD and a body mass index ≥25 kg/m^2^ from a health examination center were randomized into an intervention group using a mobile app with partial meal replacements or a control group receiving standard educational materials. Liver enzyme levels, lipid profiles, and anthropometric measurements were assessed at baseline and after 4 weeks. Twenty-five participants in the intervention group and 24 in the control group completed the trial. Significant reductions were observed in the intervention group for alanine aminotransferase (−28.32 versus [vs.] −10.67, *p* = 0.006) and gamma-glutamyl transferase (−27.76 vs. 2.79, *p* = 0.014). No significant changes in aspartate aminotransferase, body weight, or waist circumference were noted in the intervention group. Four weeks of mobile lifestyle intervention incorporating partial meal replacements improved liver enzyme profiles in patients with MASLD. This strategy demonstrated the potential for mitigating elevated liver enzyme levels without altering body weight or waist circumference. Comprehensive and longer-term research is needed to substantiate and elaborate these preliminary outcomes.

## 1. Introduction

As obesity rate increase, the prevalence of metabolic dysfunction-associated steatotic liver disease (MASLD) has rapidly increased worldwide. In Korea, it affects an estimated 20–30% of the population [1], and globally, 25% of the population and 90% of individuals with obesity have some degree of MASLD [2]. MASLD is a clinical syndrome characterized by excess lipid storage in hepatocytes and is the most common chronic liver disease worldwide [3]. While generally benign, MASLD can progress to metabolic dysfunction-associated steatohepatitis (MASH), leading to advanced liver diseases such as cirrhosis or hepatocellular carcinoma. MASLD is also an independent risk factor for metabolic disorders including cardiovascular diseases, obesity, diabetes, metabolic syndrome, and hyperlipidemia. Patients with MASLD have a 1.6-fold higher mortality rate than the general population, with cardiovascular diseases being the leading cause of death [4,5]. 

Almost all clinical practice guidelines, including the Korean Association for the Study of Liver, the European Association for the Study of Liver, European Association for the Study of Diabetes, European Association for the Study of Obesity, and the American Association for the Study of Liver Diseases, emphasize lifestyle modifications with a 5–10% reduction in weight as the cornerstone and crucial component of MASLD management [1,5].

Recently, health-related smartphone applications have increased, with lifestyle improvement programs related to blood glucose or diabetes being the most prevalent [6]. Digital healthcare offers freedom in terms of time, space, and financial constraints compared with traditional face-to-face therapy for lifestyle improvement education [7]. It offers personalized guidance, tracks structured interventions, and has the potential to enhance patient engagement and adherence to treatment protocols outside traditional healthcare settings [8]. Furthermore, in most mobile healthcare studies targeting chronic diseases, similar or improved effects (such as weight loss, reduction in waist circumference, and increased vegetable consumption in the diet) were observed compared with conventional conservative treatment, with no major side effects reported [9]. Meal replacement, which involves substituting conventional meals with portion-controlled nutrient-dense products, facilitates significant weight loss and improves liver marker levels in individuals with MASLD [10]. Recent meta-analyses revealed that meal replacement can significantly reduce body weight, body mass index, low-density lipoprotein cholesterol, non-high-density lipoprotein cholesterol, systolic blood pressure, and glycated hemoglobin (HbA1c) levels in individuals with pre-diabetes and features of metabolic syndrome [11]. These structured meal plans, combined with behavioral counseling through digital healthcare apps, can help patients achieve and maintain the recommended weight-loss targets for disease management. 

From a research perspective, it is essential to evaluate the effectiveness of digital health intervention and meal replacements separately. However, in real-world settings, it is also meaningful to examine the combined effect of these two different interventions, just like managing blood pressure using combination treatment.

To our knowledge, there is a scarcity of research examining the effects of integrating mobile apps with meal replacement in specialized lifestyle modification programs for patients with MASLD, and few randomized controlled trials have been published to date. While individual studies have explored the efficacy of meal replacement or mobile app interventions for improving MASLD [12,13], none have combined these two approaches to provide a comprehensive lifestyle modification intervention for the management of MASLD. 

Our hypotheses were as follows: (1) the mobile lifestyle intervention combined with partial meal replacements will improve liver enzyme levels in patients with MASLD, and (2) this effect may be due to more significant weight loss. 

Therefore, this study was to evaluate the effects of the mobile lifestyle intervention combined with partial meal replacement on the improvement of liver enzymes, body weight, waist circumference and other cardiometabolic parameters in patients with MASLD. By doing so, we aim to explore the potential of this new therapeutic approach in managing MASLD. 

## 2. Materials and Methods

This was a part of the 2023 Lifestyle digital leading service project led by Korean National IT Industry Promotion Agency. Given the limited duration of the project, we decided on a 4-week randomized controlled trial of mobile lifestyle intervention combined with partial meal replacement to improve liver enzymes. 

### 2.1. Study Design and Participants

This exploratory, short-term, randomized controlled trial was conducted between July 2023 and January 2024 at a secondary hospital in South Korea. Participants were randomly allocated in a 1:1 ratio to either the intervention group (using partial meal replacements with mobile coaching) or the control group (receiving usual care with educational material). Individuals with MASLD were recruited from a health promotion center, family medicine department clinic, or gastroenterology department clinic at the hospital. Patients with MASLD were invited to participate in this trial. The inclusion criteria were as follows: (1) body mass index (BMI) of ≥25 kg/m^2^, (2) those aged between 19 and 65 years, (3) those who had undergone abdominal ultrasonography within 1 month prior to screening and were observed to have hepatic steatosis, (4) alanine aminotransferase (ALT) levels between 40 and 200 with an aspartate aminotransferase (AST)/ALT ratio <1 on liver function tests, (5) owned a smartphone, and (6) ability to provide voluntary consent.

Participants were excluded if they had hepatitis B or C, consumed a significant amount of alcohol (defined as 140 g per week for women and 210 g per week for men), took medications known to induce fatty liver (such as steroids, amiodarone, methotrexate, tamoxifen, valproic acid or herbal/dietary supplements affecting liver function within the previous 6 months), had severe cardiac, pulmonary, or underlying conditions (such as congestive heart failure, ischemic heart disease, third-degree atrioventricular block, or chronic obstructive pulmonary disease) or were on oral hypoglycemic agents or insulin therapy for diabetes. All the participants provided written informed consent. The research was conducted according to the protocol approved by the Institutional Review Board of Seoul Bumin Hospital (IRB No. 2023-06-017-001) and adhered to the Declaration of Helsinki (ISCRTN registration number 16782046). 

Participants were stratified by sex, age (<60 years or ≥60 years), and BMI (<30 kg/m^2^ or ≥30 kg/m^2^) and then assigned to either the waitlist (control group) or the intervention group using sealed envelopes in a 1:1 allocation ratio. The randomization process was managed by a research assistant who was not involved in the interventions. 

### 2.2. Intervention Program 

The intervention included a 4-week program with partial high-protein meal replacement and lifestyle modification through a mobile app (Dr. Coach). High-protein meal replacements were provided by Bionutrion Corp. (Cheonan, Republic of Korea). Each sachet contained 20 g of isolated soy protein with an amino acid score of 127, 14 g of carbohydrates, and multivitamins, including vitamin D, totaling 150 kcal. These protein-supplemented partial meal replacements were recommended for consumption as dinner. Each sachet was mixed with 200 mL of low-fat milk, fat-free milk, or soymilk (approximately 5–6 g of protein) to yield 25 g protein and 240 kcal per serving. Participants in the intervention group received a unique code number for logging into the Dr. Coach app after learning how to use it and registering on the spot. 

The Dr. Coach app, developed by Bionutrion Corp., is designed to effectively manage weight loss through comprehensive lifestyle interventions with nutritionist-led mobile coaching. The app is available in app stores and Google Play stores with free access to its basic features. Study participants were granted access to the full features, including links to electronic medical record systems or coaching. This application integrates EZCareTech’s electronic medical record system, allowing it to retrieve diagnoses, prescribed medications, and laboratory results with the participants’ consent. The app’s features are broadly categorized into three main functions: (1) severity assessment using fibrosis-4 index (FIB-4) [14] with age, AST, ALT, and platelet and the non-alcoholic fatty liver disease fibrosis score [15] with the previous FIB-4 variables, sex, albumin and BMI, (2) personalized nutrition prescription using electronic medical record-linked data for consideration of 30 underlying diseases including diabetes, hypertension, dyslipidemia, gout or osteoporosis, and (3) self-monitoring of weight, diet, activity levels, emotions, and sleep, along with mobile coaching to offer support and feedback. Figure 1 illustrates the main features and functionalities of the Dr. Coach app used in the intervention. The app’s interface includes sections for severity assessment using the fibrosis-4 index and non-alcoholic fatty liver disease fibrosis score, personalized nutrition prescription, and self-monitoring of weight, diet, activity levels, emotions, and sleep. The mobile coaching feature provides daily support and feedback, helping participants adhere to their lifestyle modification goals.

Mobile coaching is implemented in a chat format 5 days a week and includes the following functions, as suggested in the app behavior change scale [16]: (1) knowledge education: definition, disease course, and prognosis of MASLD, nutritional interventions, and physical activity guidelines; (2) goal setting and planning: specific, measurable, achievable, relevant, and time-limited goal setting and planning; (3) monitoring and feedback: self-monitoring of weight, daily meals consumed, physical activities, sleep, emotions if needed, and providing users with feedback and support; and (4) helping actions: reminders for planned meals or activities, providing alternatives for barriers, encouraging positive habit formation, and restructuring. Participants in the control group received standard care, consisting of brief education provided by healthcare providers. 

### 2.3. Study Outcomes and Measurements

The primary outcome was a reduction in liver enzymes, including AST, ALT, gamma-glutamyl transferase (GGT), and alkaline phosphatase, at 4 weeks between the two groups. Secondary outcomes included changes in body weight, waist circumference, body fat percent, systolic blood pressure, diastolic blood pressure, triglyceride, high-density cholesterol, low-density cholesterol, and fasting blood glucose levels.

Body composition, including body weight, was measured automatically using bioelectrical impedance analysis (Accuniq BC 720, Venlo, The Netherlands) at each visit, with the participants in a standing position with light clothing and fasting. Waist circumference was measured barefoot to the nearest 0.1 cm at the midline between the lowest rib margin and the anterior superior iliac spine while standing after normal expiration. Systolic and diastolic blood pressures were measured in the seated position using an automated sphygmomanometer (two measurements and an average were obtained). The study outcomes were measured at each visit, except for height measurements. 

After a 10 h fast, blood samples were tested for white blood cell count, hemoglobin, platelet count, fasting glucose, HbA1c, total cholesterol, high-density cholesterol, triglycerides, and low-density cholesterol. Levels of AST, ALT, GGT, alkaline phosphatase, albumin, bilirubin, uric acid, and eGFR were calculated using the creatinine-based Chronic Kidney Disease Epidemiology Collaboration equation [17]. Serological tests for hepatitis B surface antigen and hepatitis B surface antibody and enzyme-linked immunosorbent assay for hepatitis C were performed for all individuals to ensure the absence of hepatitis. 

The Korean version of the obesity-related quality of life scale [18] and a dietary pattern evaluation tool [19] were conducted at randomization and the end of the study period.

### 2.4. Statistical Analysis

Since mobile app-based coaching and high-protein partial meal replacement have not been tried before, and owing to the project period’s limitation, the study was designed as an exploratory pilot study aiming to enroll 60 participants, with 30 in each group. The primary endpoints included a reduction in liver enzyme levels after 4 weeks of the intervention program between the two groups. Between-group differences were assessed using two-sample *t*-tests or Mann–Whitney U tests for continuous variables and the chi-square test or Fisher’s exact test for categorical variables. Changes during the month were analyzed using paired *t*-tests, or Wilcoxon signed-rank tests for continuous variables, and McNemar’s test for categorical variables within each group. All statistical analyses were performed using SAS statistical software (version 9.4; SAS Institute, Cary, NC, USA), with the significance level set at 0.05. 

## 3. Results

### 3.1. Participants’ Characteristics

Data on the number of participants screened, excluded, randomized, and included in the analysis are shown in Figure 2. Sixty participants were randomized to the intervention (n = 30) or control groups (n = 30), among whom 49 completed the study. Five participants in the intervention group and six in the control group dropped out of the study. The participants had a mean age of 41.4 (±6.4) years, a mean BMI of 29.7 ± 4.2 kg/m^2^, and 94% were male. The main clinical characteristics of the study population are summarized in Table 1. There were no significant differences in demographic and anthropometric measurements between the intervention and control groups. The participants’ mean AST, ALT, and GGT were 42.9 (±14.5) IU/dL, 71.9 (±33.8) IU/dL, and 66.4 (±4.07) IU/dL with no significant differences between groups. There were no significant differences in cardiometabolic risk factors (blood pressure, cholesterol levels, glucose, and HbA1c), quality of life (QOL), and control groups (Table 1).

### 3.2. Changes in Liver Enzymes and Body Weight

As shown in Table 2 and Figure 3, over 4 weeks, the intervention group showed a significant decrease in ALT (−28.32 ± 25.93 IU/L versus [vs.] −10.67 ± 15.54, *p* = 0.006) and GGT (−27.76 ± 25.02 IU/L vs. 2.79 ± 54.22, *p* = 0.014) compared with the control group. On completion of the study, the participants in the intervention group lost an average of −2.93 ± 2.58 kg (3.2% reduction in baseline weight), whereas those in control group lost an average of −2.74 ± 1.54 kg (3.1% reduction in baseline weight) (*p* = 0.762) There was also a marginally significant decrease in AST (−13.28 ± 20.30 IU/L vs. −4.25 ± 12.98, *p* = 0.071). Figure 3 shows the changes in liver enzyme levels (AST, ALT, GGT) between the intervention and control groups over the 4-week study period. Notably, the intervention group achieved a significantly greater reduction in percent body fat than the control group, with decreases of −1.46 ± 1.31 vs. −0.48 ± 1.48% (*p* = 0.016), respectively. 

### 3.3. Adverse Events 

Three participants in the intervention group (two upper respiratory infections and one allergy symptom) and four in the control group (two upper respiratory infections, one allergy, and one abdominal discomfort) reported adverse events during the study period. These symptoms were mild, transient, and improved voluntarily. 

## 4. Discussion

This study explored the potential benefits and feasibility of short, individualized, app-based lifestyle interventions along with high-protein meal replacement for patients with MASLD. In the intervention group, which received meal replacement and app-based lifestyle modification coaching for 4 weeks, showed significant reductions in ALT and GGT enzyme levels compared with the control group. Despite minimal differences in weight and waist circumference between the groups, the intervention group demonstrated a more statistically and clinically significant reduction in liver enzyme levels. While previous research has examined interventions such as meal replacement or lifestyle modifications [13,20], our new approach suggests the potential for combining high-protein meal replacements with mobile lifestyle interventions as a therapeutic modality for MASLD. 

Normalization of liver biochemical parameters usually reflects the histological response to MASLD [21]. Elevated ALT and AST levels, in the absence of other liver diseases, support a diagnosis of MASLD and are present in approximately 50% of individuals with simple steatosis and 80% of those with MASH. Additionally, an AST/ALT ratio of <1 is frequently observed in MASLD and supports its diagnosis [22]. The observed reductions in ALT, GGT levels, and weight loss indicated a marked improvement in hepatic health and metabolic status among participants with MASLD in the intervention group. 

Dietary modification plays a crucial role in managing MASLD, with reducing total energy intake being the most critical aspect [23]. Research has demonstrated that weight-loss programs tailored for patients with MASLD can achieve significant outcomes, including a weight loss of 5–10% and the normalization of liver enzymes [24]. The Mediterranean diet is highly regarded for treating MASLD and is currently recommended by the KASL, EASL-EASD-EASO, and AASLD. However, other diets, such as low-carbohydrate, high-protein, Paleolithic, ketogenic, plant-based diets, and intermittent fasting protocols, seem promising for patients with MASLD [25]. 

Adherence to these dietary interventions is crucial for managing MASLD but remains a significant challenge. Poor adherence can lead to disease progression, increased liver damage [26], worsened metabolic complications [27], and impaired QOL, highlighting the importance of promoting adherence through patient education, social support, and individualized dietary plans. Recent systematic reviews and meta-analyses support the use of meal replacements for weight loss in individuals with pre-diabetes or features of metabolic syndrome. Furthermore, they significantly reduce weight, BMI, body fat, fasting blood glucose, and HbA1c levels in patients with type 2 diabetes and obesity [28] and are effective in improving MASLD [29]. Moreover, meal replacements have the advantage of a clearly labeled calorie content, making it easier to control calorie intake. They generally provide balanced nutrients and are convenient to prepare and consume [30]. In particular, high-protein meal replacements can help reduce body fat, preserve muscle mass, and increase satiety [31]. Increased dietary protein intake in the background of a hypocaloric diet has been shown to improve serum lipid levels, glucose homeostasis, and liver enzymes in patients with MASLD, independent of changes in BMI or body fat mass [32]. Therefore, in this study, the high-protein diet replacement intervention lasted only for 4 weeks; it is believed to have contributed to the improvement in MASLD without significant differences in body weight or waist circumference. Using meal replacement for one or two meals a day offers a practical alternative for Koreans, eliminating the need to adhere to complex diets such as the Mediterranean, low-carbohydrate, or ketogenic diets, which may be challenging to adopt. 

A recent meta-analysis of weight-loss interventions in adults with chronic diseases found a significant weight reduction (2.5 kg) among users in the mobile app group compared to the control group [33]. Furthermore, research by Lim et al. revealed that patients with MASLD who received advice from nutritionists regarding dietary and lifestyle modifications through a mobile app achieved more than a five-fold increase in weight loss of >5% and improvements in liver enzymes after 6 months [13]. Another study that examined lifestyle changes in patients with MASLD reported an average weight loss of 3.4 kg from baseline body weight at 6 months through an online remote coaching method [34]. Our findings contribute to the existing evidence that mobile apps featuring self-monitoring and remote coaching can significantly reduce the weight of patients with MASLD. The mobile text messaging approach has also been effective in facilitating changes in health behavior [35], leading to weight loss and other beneficial outcomes. This method, which is used in digital coaching for this study, improves communication speed and ease. 

For the treatment of MASLD, a combination of meal replacement and app-based coaching programs may be more effective. While meal replacement aids in weight loss by reducing the cognitive burden associated with meal preparation and consideration of caloric and macronutrient composition, long-term success necessitates continuous lifestyle changes supported by health coaching through real-time feedback and motivation [36]. App-based coaching programs provide personalized diet and exercise plans, thereby enhancing therapeutic effects. In addition, these programs offer continuous education and support for nutrition and healthy habits. Their convenience and accessibility enable ongoing monitoring and timely interventions, integrating seamlessly into busy lifestyles [37,38]. Therefore, by comprehensively addressing diet and lifestyle, the synergy between meal replacement and app-based coaching optimizes MASLD treatment, making it more effective and sustainable.

This study has several strengths. One notable strength is the pioneering approach of integrating app-based lifestyle modifications with meal replacement for the first time. It showed the potential for investigating the combined effects of a mobile app using evidence-based strategies and a high-protein meal replacement on improving physiological and biochemical markers associated with patients with MASLD beyond weight loss. Furthermore, this program could be applicable not only to other chronic conditions such as diabetes and cardiovascular diseases but also to a wider population. However, more rigorous and detailed studies with stricter designs are required to determine which specific components contributed to the improvements and at what stages these changes occurred. 

This study excluded users without smartphones and illiterate users. Smartphone ownership rates are increasing but still depend on educational level and household income [39]. Therefore, the proportion of individuals with lower socioeconomic status may have been lower in this clinical trial. The short study duration may not have captured the long-term effects or sustainability of the intervention. Our study also lacks generalizability to a broader population as it recruited patients from a single healthcare institution. Finally, a limitation of this study is the uncertainty regarding whether the coaching program or meal replacement was effective in improving liver function. In addition to employing more rigorous study designs, it is necessary to evaluate cost–benefit analysis and acceptability across different age and sex groups. 

## 5. Conclusions

Our pilot study is the first to present the potential of a comprehensive treatment approach for MASLD, integrating meal replacement therapy with a lifestyle modification mobile app and coaching. 

Significant improvements in liver enzyme levels (ALT and GGT) were observed in the intervention group.The intervention did not cause significant changes in body weight or waist circumference.The combination of mobile lifestyle intervention and high-protein meal replacement shows potential as a therapeutic approach for MASLD.Further long-term studies are needed to substantiate and elaborate on these preliminary outcomes.

## Figures and Tables

**Figure 1 nutrients-16-02254-f001:**
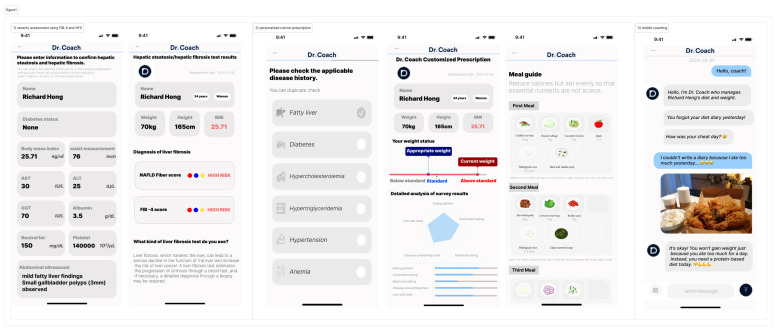
Dr. Coach App and its function.

**Figure 2 nutrients-16-02254-f002:**
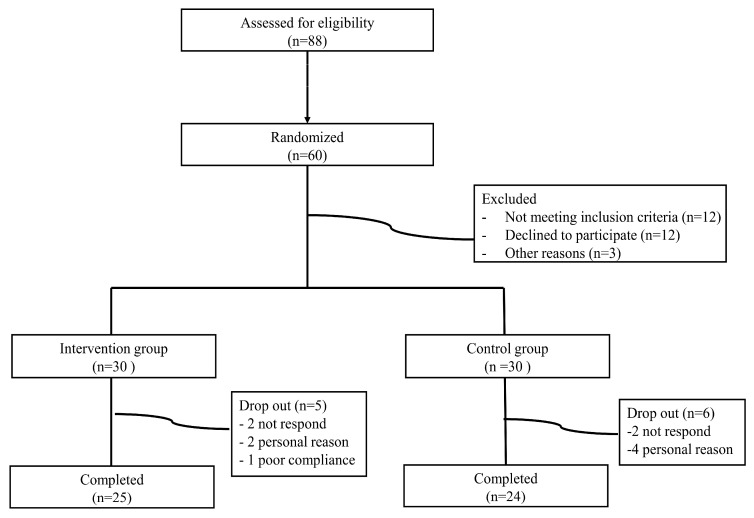
Flow diagram of participant selection and allocation in the study.

**Figure 3 nutrients-16-02254-f003:**
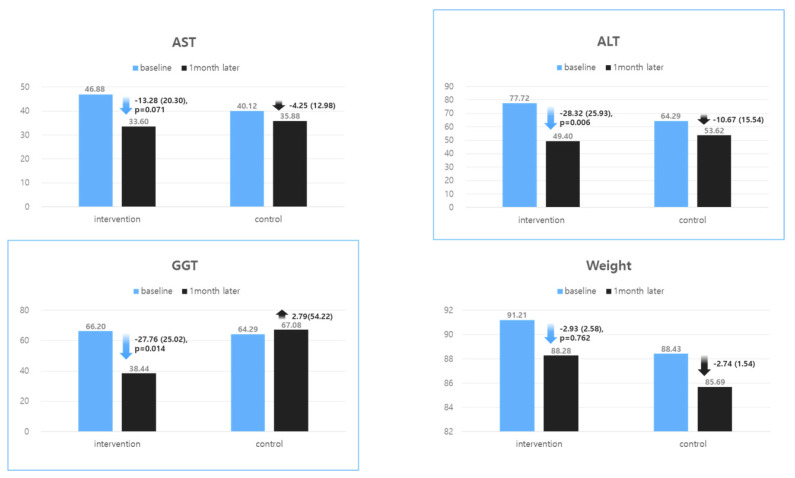
Changes in liver enzymes in the intervention and control groups.

**Table 1 nutrients-16-02254-t001:** Baseline characteristics.

	Intervention (N = 25)	Control (N = 24)	*p*-Value
Age (years), mean (SD)	41.32 (6.74)	43.50 (6.67)	0.261
Male sex (%)	25 (100.0)	21 (87.5)	0.068
Body weight (kg), mean (SD)	91.21 (12.37)	88.43 (15.35)	0.489
BMI (kg/m^2^), mean (SD)	30.04 (4.48)	30.24 (3.70)	0.866
Waist circumference (cm), mean (SD)	102.36 (10.64)	101.73 (12.51)	0.850
Body fat percent (%), mean (SD)	30.30 (5.58)	31.58 (5.74)	0.433
Systolic blood pressure (mmHg), mean (SD)	131.32 (15.23)	133.42 (19.00)	0.671
Diastolic blood pressure (mmHg), mean (SD)	81.38 (13.33)	80.56 (8.09)	0.796
AST (IU/L), mean (SD)	46.88 (20.00)	40.12 (11.38)	0.155
ALT (IU/L), mean (SD)	77.72 (38.20)	64.29 (27.21)	0.165
GGT (IU/L), mean (SD)	66.20 (43.08)	64.29 (45.32)	0.881
Alkaline phosphatase (IU/L), mean (SD)	69.08 (22.10)	73.25 (20.04)	0.497
Albumin (IU/L), mean (SD)	4.64 (0.18)	4.74 (0.15)	0.038
Total cholesterol (mmol/L), mean (SD)	204.88 (42.99)	199.46 (39.66)	0.649
HDL-cholesterol (mmol/L), mean (SD)	43.60 (7.92)	45.58 (9.66)	0.435
LDL-cholesterol (mmol/L), mean (SD)	122.74 (32.79)	121.19 (35.24)	0.874
Triglycerides (mmol/L), mean (SD)	192.72 (116.37)	163.42 (87.46)	0.326
Fasting glucose (mmol/L), mean (SD)	96.04 (6.72)	97.46 (13.75)	0.646
WBC (×10^3^/µL), mean (SD)	6.67 (1.93)	6.79 (1.93)	0.83
Hb (g/dL), mean (SD)	16.34 (0.71)	15.81 (1.24)	0.075
Platelet (×10^3^/µL), mean (SD)	251.08 (53.52)	265.00 (57.23)	0.383
Smoking status			0.91
Non-smoker	11 (44.0)	12 (46.2)	
Ex-smoker	9 (36.0)	10 (38.5)	
Current smoker	5 (20.0)	4 (15.4)	
Household income, N (%)			0.19
<300,000	2 (8.0)	0 (0.0)	
<500,000	4 (16.0)	10 (38.5)	
<800,000	7 (28.0)	9 (34.6)	
<100,000	5 (20.0)	3 (11.5)	
>100,000	7 (28.0)	4 (15.4)	
Educational level, N (%)			0.492
Less than high school	1 (4.0)	0 (0.0)	
High school	1 (4.0)	0 (0.0)	
College or more	19 (76.0)	20 (76.9)	
Graduate or more	4 (16.0)	6 (23.1)	
Subjective health, N (%)			0.727
Very good	2 (8.0)	2 (8.3)	
Good	19 (76.0)	16 (66.7)	
Bad	4 (16.0)	6 (25.0)	
Alcohol consumption, N (%)			0.317
None	3 (12.0)	1 (4.2)	
Moderate	22 (88.0)	23 (95.8)	
Hypertension, N (%)			0.928
Yes	7 (28.0)	7 (29.2)	
No	18 (72.0)	17 (70.8)	
Dyslipidemia, N (%)			0.24
Yes	3 (12.0)	6 (25.0)	
No	22 (88.0)	18 (75.0)	
Gout, N (%)			0.302
Yes	0 (0.0)	1 (4.2)	
No	24 (96.0)	23 (95.8)	
Thyroid disease, N (%)			0.976
Yes	1 (4.0)	1 (4.2)	
No	24 (96.0)	23 (95.8)	

All such values are presented as mean ± standard deviation. Student’s *t*-test for continuous variables and the chi-square test for categorical variables were used to compare differences between the groups. SD, standard deviation; BMI, body mass index; ALT, alanine aminotransferase; GGT, gamma-glutamyl transferase; LDL, low-density lipoprotein; WBC, white blood cell; Hb, hemoglobin.

**Table 2 nutrients-16-02254-t002:** Mean (SD) changes in anthropometric, biochemical, and clinical outcomes in patients with MASLD at 4 weeks.

	Intervention (N = 25)		Control (N = 24)		*p*-Value
	Changes from Baseline	Within Group *p*-Value	Changes from Baseline	Within Group *p*-Value	
Weight (kg), mean (SD)	−2.93 (2.58)	0.405	−2.74 (1.54)	0.537	0.762
BMI (kg/m^2^), mean (SD)	−0.98 (0.87)	0.43	−0.94 (0.51)	0.377	0.843
Body fat percent (%)	−1.36 (1.34)	0.393	−0.70 (1.50)	0.674	0.115
Waist circumference (cm), mean (SD)	−2.73 (2.38)	0.357	−2.22 (2.08)	0.54	0.428
Systolic blood pressure (mmHg), mean (SD)	0.64 (17.21)	0.872	−1.71 (18.67)	0.75	0.649
Diastolic blood pressure (mmHg), mean (SD)	−0.08 (9.69)	0.974	−1.25 (12.27)	0.726	0.712
AST (IU/L), mean (SD)	−13.28 (20.30)	0.009	−4.25 (12.98)	0.34	0.071
ALT (IU/L), mean (SD)	−28.32 (25.93)	0.002	−10.67 (15.54)	0.196	0.006
GGT (IU/L), mean (SD)	−27.76 (25.02)	0.007	2.79 (54.22)	0.888	0.014
Alkaline phosphatase (IU/L), mean (SD)	−0.96 (5.81)	0.926	1.46 (7.49)	0.813	0.218
Albumin (IU/L), mean (SD)	−0.06 (0.15)	0.286	0.07 (0.21)	0.212	0.017
Total cholesterol (mmol/L), mean (SD)	−12.92 (27.84)	0.282	−5.54 (22.49)	0.656	0.314
HDL-cholesterol (mmol/L), mean (SD)	−1.20 (6.18)	0.601	1.04 (4.40)	0.694	0.152
LDL-cholesterol (mmol/L), mean (SD)	−2.95 (38.19)	0.776	−1.39 (21.95)	0.899	0.862
Triglycerides (mmol/L), mean (SD)	−43.84 (131.72)	0.131	−25.96 (65.92)	0.221	0.553
Fasting glucose (mmol/L), mean (SD)	−1.52 (8.02)	0.475	0.17 (9.75)	0.961	0.511
WBC (×10^3^/µL), mean (SD)	−0.25 (1.42)	0.604	0.20 (1.13)	0.72	0.228
Hb (g/dL), mean (SD)	−0.14 (0.51)	0.484	0.03 (0.69)	0.943	0.346
Platelet (×10^3^/µL), mean (SD)	−14.20 (26.18)	0.323	3.17 (30.61)	0.852	0.038

Linear mixed-effect model adjusted to analyze the differences within each group. SD, standard deviation; BMI, body mass index; AST, aspartate aminotransferase; ALT, alanine aminotransferase; GGT, gamma-glutamyl transferase; HDL, high-density lipoprotein; LDL, low-density lipoprotein; WBC, white blood cell; Hb, hemoglobin.

## Data Availability

The data presented in this study are available on request from the corresponding author due to restrictions in privacy.

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
