# Peer review of "The Effect of Mobile Lifestyle Intervention Combined with High-Protein Meal Replacement on Liver Function in Patients with Metabolic Dysfunction-Associated Steatotic Liver Disease: A Pilot Randomized Controlled Trial"

_nutrients, 2024, doi:10.3390/nu16142254_

Round 1

Reviewer 1 Report

Comments and Suggestions for Authors

The study is well written and touches on a hot topic.

Here are my suggestions:

The study is large and ambitious. I suggest in the introduction to include some hypotheses of what you expect to find. Furthermore, the aim of a single farce rich in content does not honor the wide-ranging study.

Figure 1 is illegible and should also be commented on in detail in the body of the manuscript.

Avoid paragraphs of just a few lines (such as paragraph 2.2.1)

The methods are divided into themes organized into paragraphs. It is suggested to insert a premise before the paragraphs and perhaps a flow chart that helps to summarize the design of the study.

The figure 2 caption is not clear. It does not refer to the study design but to selection of participants. Please clarify.

Figure 3 is with low resolution and must be described in detail.

The conclusions must be expanded emphatizing the key findings. Use bullet points..

Author Response

Thank you very much for your positive evaluation of our study and your valuable suggestions. We are committed to enhancing our manuscript and have carefully considered your recommendations.

Comments1: Include some hypotheses in the introduction to clearly outline what the study aims to find.

Response1: We agree that including hypotheses would provide a clearer framework for the study's aims and expectations.

We removed the previous overly broad description of MASLD and its effect on overall morbidity, cost or mortality. Instead we focused more on the rationale for integrating digital health interventions and meal replacements, as well as  the specific hypothesis of this study.

We added  the following hypotheses to the introduction as in line 82-84.

Comment 2: Separate the aim into a single, rich in content paragraph to better reflect the wide-ranging scope of the study.

Response2: We revised the introduction to clearly state the study's objectives in a standalone paragraph. The revised paragraph is as follows in line 85-89

"The objective of this study is to evaluate the effects of a mobile lifestyle intervention combined with partial high-protein meal replacement on liver enzyme levels, body weight, waist circumference, and other cardiometabolic parameters in patients with MASLD. By doing so, we aim to explore the potential of this new therapeutic approach in managing MASLD."

Comment 3 : Figure 1 is illegible and should also be commented on in detail in the body of the manuscript.

Response 3: We acknowledge that the current version of Figure 1 is difficult to read. We provide a higher-resolution image to ensure legibility. Additionally, we included a detailed description of Figure 1 in the manuscript body to enhance understanding. The revised text will include:

  • "Figure 1 illustrates the main features and functionalities of the Dr. Coach app used in the intervention. The app's interface includes sections for severity assessment using the fibrosis-4 index and non-alcoholic fatty liver disease fibrosis score, personalized nutrition prescription, and self-monitoring of weight, diet, activity levels, emotions, and sleep. The mobile coaching feature provides daily support and feedback, helping participants adhere to their lifestyle modification goals."

Comments 4: Avoid paragraphs of just a few lines (such as paragraph 2.2.1)

Response4: We removed the paragraph 2.2.1 to ensure that short paragraphs are merged or expanded to provide more comprehensive information.

Comments 5:The methods are divided into themes organized into paragraphs. It is suggested to insert a premise before the paragraphs and perhaps a flow chart that helps to summarize the design of the study.

Response 5: We included an overview of the study design in the methods sectionas in line 91-94. Given the short 4-week trial duration without any interim visits, we did not include a flow chart.

Comments 6: The figure 2 caption is not clear. It does not refer to the study design but to selection of participants. Please clarify.

Response 6: We revised  the caption of Figure 2 to clearly indicate that it refers to the selection and flow of participants throughout the study as in page 6.

"Figure 2. Flow diagram of participant selection and allocation in the study."

Comments 7: Figure 3 is with low resolution and must be described in detail.

Response 7: We replaced Figure 3 with a higher-resolution image and include a detailed description in the manuscript as in page 10.

"Figure 3 shows the changes in liver enzyme levels (AST, ALT, GGT) between the intervention and control groups over the 4-week study period.”

Comments 8:The conclusions must be expanded emphatizing the key findings. Use bullet points..

Response 8: We will expand the conclusions section to emphasize the key findings of the study. The revised conclusions will be presented as bullet points for clarity:

  • Significant improvements in liver enzyme levels (ALT and GGT) were observed in the intervention group.
  • The intervention did not cause significant changes in body weight or waist circumference.
  • The combination of mobile lifestyle intervention and high-protein meal replacement shows potential as a therapeutic approach for MASLD.
  • Further long-term studies are needed to substantiate and elaborate on these preliminary outcomes.

Reviewer 2 Report

Comments and Suggestions for Authors

The manuscript titled: The Effect of Mobile Lifestyle Intervention Combined with High-Protein Meal Replacement on Liver Function in Patients with Metabolic Dysfunction-Associated Steatotic Liver Disease: A Pilot Randomized Controlled Trial presents short-term mobile interventions coupled with partial meal replacement in patients with MASLD.

1.      How often is MASLD diagnosed in Korea compared to worldwide appearance?

2.      How did you calculate the amount of groups needed for this study? Are 24/25 subjects enough to do such an assessment?

3.      Why did you choose to change only one meal?

4.      Did you monitor the patient personally – e.g., call every week to check if everything is under is everything understandable?

5.      Did you monitor ESR or CRP? These parameters reflect the inflammation, which is increased in overweight and obese.

6.      What is the novelty of your study?

Author Response

Thank you very much for your positive evaluation of our study and your valuable suggestions. We are committed to enhancing our manuscript and have carefully considered your recommendations.

Comments 1: How often is MASLD diagnosed in Korea compared to worldwide appearance?

Response 1: The prevalence of MASLD in Korea is estimated to be around 20-30% of the population, which is comparable to the global prevalence of approximately 25%​​. This high prevalence underscores the importance of effective management strategies such as the one explored in our study. This part was introduced in line 35-37.

Comments 2:  How did you calculate the amount of groups needed for this study? Are 24/25 subjects enough to do such an assessment?

Response 2: Our study was designed as an exploratory pilot trial, aiming to evaluate the feasibility and preliminary effects of the intervention. The sample size of 60 participants, with 30 in each group, was determined based on practical considerations and the exploratory nature of the study. While a larger sample size would provide more robust statistical power, the current sample size allows for initial insights and will inform the design of future larger-scale studies. The observed significant changes in liver enzyme levels within the intervention group suggest that the sample size, though modest, was sufficient to detect meaningful effects in this pilot phase.

Comments 3:     Why did you choose to change only one meal?

Response 3: We chose to change only one meal given the short intervention period of only 4 weeks. This approach aims to enhance adherence to the intervention and minimize the burden on participants. By focusing on a single high-protein meal replacement, we aimed to create a manageable and sustainable dietary change that participants could realistically incorporate into their daily routines. This approach also allowed us to isolate the effects of the meal replacement more effectively.

Comments 4:      Did you monitor the patient personally – e.g., call every week to check if everything is under is everything understandable?

Response 4: Yes, patient monitoring and support were integral components of the intervention. Participants in the intervention group received personalized coaching and support through the Dr. Coach mobile app. The app provided daily feedback and support, and participants had access to a nutritionist for any questions or concerns. Additionally, participants received weekly check-ins via the app to ensure adherence and address any issues. This digital approach allowed for continuous monitoring and support without the need for frequent in-person visits.

Comments 5: Did you monitor ESR or CRP? These parameters reflect the inflammation, which is increased in overweight and obese.

Response 5: In this pilot study, we did not include erythrocyte sedimentation rate (ESR) or C-reactive protein (CRP) as part of our monitored parameters. The primary focus was on liver enzyme levels, body weight, and other cardiometabolic parameters. However, we acknowledge the importance of these inflammatory markers and plan to include them in future studies to provide a more comprehensive assessment of the intervention's impact on inflammation associated with MASLD and obesity.

Comments 6:      What is the novelty of your study?

Response 6: The novelty of our study lies in the integration of a mobile lifestyle intervention with high-protein meal replacement specifically targeting patients with MASLD. While previous studies have examined dietary interventions or mobile health applications separately, our study is among the first to combine these approaches to explore their synergistic effects on liver function. Additionally, our use of a digital health platform for personalized coaching and monitoring represents an innovative approach to enhancing adherence and engagement in lifestyle modification programs. These aspects collectively highlight the potential of combining technological and nutritional strategies to manage MASLD effectively.